# Exploring the 4IR Drivers for Sustainable Residential Building Delivery from Social Work Residential Perspective—A Structural Equation Modelling Approach

Ayodeji Emmanuel Oke [1,2,*], Ahmed Farouk Kineber [3,*], Oludolapo Ibrahim Olanrewaju [4], Olayinka Omole [5], Paramjit Singh Jamir Singh [2,*], Mohamad Shaharudin Samsurijan [2] and Rosfaraliza Azura Ramli [6]

1   Department of Quantity Surveying, Federal University of Technology Akure, Akure 340110, Nigeria
2   School of Social Sciences, Universiti Sains Malaysia, Penang 11800, Malaysia
3   Department of Civil Engineering, College of Engineering in Al-Kharj, Prince Sattam Bin Abdulaziz University, Al-Kharj 11942, Saudi Arabia
4   Wellington School of Architecture, Victoria University of Wellington, Wellington 6140, New Zealand
5   Irene Construction Science Division, Christopher Gibbs College of Architecture, University of Oklahoma, Norman, OK 73019, USA
6   College of Law, Government and International Studies, School of Government, Universiti Utara Malaysia, Changlun 06050, Malaysia
*   Correspondence: emayok@gmail.com (A.E.O.); a.kineber@psau.edu.sa (A.F.K.); paramjit@usm.my (P.S.J.S.)

**Abstract:** The advent of digitalization has brought many benefits to all sectors of the economy, including construction. When fully implemented, various Fourth Industrial Revolution (4IR) tools have the potential not only to improve project planning and execution, but also to enhance project performance. This study therefore investigated the critical factors for the adoption of 4IR technologies in the construction industry, with the aim of promoting sustainable construction project delivery. The study was conducted using a questionnaire sent to experts in the construction industry. The data collected were analyzed using exploratory factor analysis (EFA) and categorized into operational, management, and demographic variables. Partial Least Square Structural Equation Modeling (PLS-SEM) was used for model development using the four groups of data. In this way, variables that were not significant to the model were identified. Judging from the analysis, there is a need for proper user training in engineering tools in the construction industry. This is one of the drivers of the adoption of 4IR in the construction industry. In addition, the professionals, contractors, authorities, and other stakeholders responsible for managing projects in the architecture, engineering, construction, and operations (AECO) industry should ensure effective coordination and collaboration between participants in the construction industry.

**Keywords:** construction innovation; digital construction; Fourth Industrial Revolution; project success; sustainable development





## 1. Introduction

Residential construction is one of the essential community conditions that ensure a healthy quality of life and well-being for residents in any country [1]. Residential buildings consume about forty percent of global power and generate up to one-third of global greenhouse gas (GHG) emissions in developed and emerging nations [2,3]. Compared to other industries of the economy, such as the manufacturing, banking, and health sectors, there has been a lack of accurate standards and efficient project management practices in the building industry [4,5]. The susceptibility of the building industry to poor management and performance due to its multifaceted nature has been acknowledged [6,7]. Over the years, the problem of project performance failure has gone untreated, and this has increased the negative impact on the building industry and the economy. Nevertheless, in an ever-changing and urbanizing world, residential allocation cannot sufficiently meet demand [8].

The world's population keeps increasing rapidly, and it recently reached over eight billion people [9]. Consequently, rapid urbanization is impeding the access of low-wage earners to affordable housing in both developing and developed countries [10]. In developing countries, it is estimated that 828 million poor people are living in slums and substandard homes. The speculation is that this figure will rise to 1.4 billion by 2020 [8,11,12]. These regions have undergone rapid development, which clearly highlights the key role of residential building in ensuring simple living [13]. As a result, all governments have prioritized affordable residential buildings by initiating several affordable residence policies [1]. Nevertheless, there is controversy about whether residential buildings are affordable for low-income earners [8].

Recently, the predominant points in defining the success of a project have been the project cost, the duration of the project and the quality requirement of the project [4]. The diverse, uncertain, and dynamic nature of the majority of construction projects has created apparent difficulty concerning meeting their initial declared objective [14]. For instance, estimates showed that 43% of developments failed to achieve all performance requirements in the year 2012 [15]. This highlighted the fact that poor project supervision is a widespread problem which contributes to the deplorable state of many countries, including Nigeria. Given that it may cause more serious issues in upcoming construction projects, this situation needs to be rectified [16]. The need for constructing "sustainable buildings" that are environmentally friendly and resource-efficient has been highlighted in the literature [17]. Wolstenholme et al. [18] further advocate for revolutionizing the building field by adopting effective and sustainable building practices. Furthermore, building professionals cannot measure the environmental influences of buildings as they accrue through construction [19]. Therefore, the Fourth Industrial Revolution can be combined with the sustainability method at the preliminary and design phases of a project. Although many research works have been carried out to categorize the critical factors disturbing the performance of building projects in Nigeria, not many studies have identified and assessed the efficacy of the Fourth Industrial Revolution principles in mitigating these factors causing poor construction project performance.

The digitalization of pertinent data, quick problem-solving, a collaborative work environment in project scheme design, in-depth building and operation, timely and adequate resource utilization, and improved quality and safety are just a few of the many advantages of adopting 4IR in the construction industry.

Although many industries have embraced these innovations in full, the building industry is still lagging [20]. Many developed countries and their construction industries have been making use of some of these innovations due to their advantages. Such countries as China, Japan, the USA, and the UK, to mention a few [21], fall within this bracket. The case is different for most African countries, especially Nigeria. The commonest and most used form of 4IR present in the Nigerian building industry is building information modeling (BIM), and even this application is yet to be fully adopted. The major concern that emanates from here is that the world is fast-moving, and many countries in Africa are being left behind. The short- and long-term implications of this issue, if it persists, are that the ways in which our industry operates will phase out. Thus, competition with international companies will gradually dry out. Therefore, effectiveness that has corroded over the past years, or in some cases, plummeted, will continue to deteriorate and diminish. The advantages that arise from the use of innovations birthed during the Fourth Industrial Revolution, which will make for the simplification of work processes, will not be enjoyed on this side of the divide. In light of this, the purpose of this study is to assess the motivations behind the adoption of 4IR advancements in construction projects and identifying the 4IR implementation drivers.

Based on our arguments, we set out the following research question for this empirical study: what are the requirements for implementing 4IR in Nigerian residential building projects? Therefore, there is a need to examine these requirements, which can be achieved by defining the drivers of 4IR [22]. Rockart [23] identifies drivers as "areas where,

if satisfactory, the results will ensure the organization's competitive success." Similarly, Chan et al. [24] and Yu et al. [25] agree that drivers may be considered as critical management preparation and action fields for ensuring success [26]. Furthermore, the drivers of 4IR present active customer support and participation [22] through decision-makers [27]. One of the first initiators of research on this topic is Romani [28]. Nonetheless, Shen and Liu [29] are credited with identifying drivers by contrasting unique practices in Hong Kong, the USA, and the UK. Despite these modest efforts, there are no data available with regard to the Nigerian construction industry. Thus, this study aims to identify the drivers of 4IR using causal inference techniques, such as structural equation modeling (SEM), in order to develop the requirements for implementing 4IR and achieving sustainability in residential building.

## 2. Drivers for 4IR Implementation

The current work culture and environment in the building industry are changing at speed globally. These changes have brought about an increase in the use of high-end innovative technologies (including the 4IR innovations), which will eventually bring forth increased efficiency, competitiveness, and effectiveness in building project management concerning its work processes and products [30,31]. Several types of studies have suggested that the construction industry, just like other industries, will adopt the use of 4IR innovations if particular factors or drivers are in place. Rapid digitization, as well as automated jobs, as part of the Fourth Industrial Revolution (4IR) will have great consequences on the career aspirations of individuals [32]. Therefore, a high level of skills will be required for upcoming digital economies, which will be driven by advanced ICT infrastructures. Digital literacy is a highly valuable skill in such economies [33]. Hence, people's ability to use and understand these sophisticated innovations will eventually facilitate their adoption. Additionally, governments and regulators must quickly become familiar with the rapidly changing 4IR landscape to provide the supportive atmosphere, protections, venture capital, and oversight necessary to educate and direct on future building projects [34]. To unleash and expand innovation in potentially revolutionary new technologies and global solutions, support and collaboration will be necessary. To avoid unexpected outcomes and safeguard public interest, consideration, public policy, and technology governance will be necessary [35]. As a result of this attention given to training and guidance, the adoption of 4IR innovations will eventually increase. The pace of technological advancement has accelerated since the start of the industrial revolution. Industrial units were initially driven by water and steam engines in the nineteenth century. Production increased in the twentieth century with the advent of electricity, and subsequently, we saw automation in the 1970s [36]. Our current position is at the pinnacle of Industry 4.0, a new phase of digital industrial technology (4IR). Using machine learning, artificial intelligence (AI), big data, the internet of things (IoT), and other technologies, cyber–physical systems can interact with one another using this fourth technology wave in the building industry. Productivity and development will increase thanks to "Industry 4.0" in ways that are not necessary for the industry's continuation and survival [36,37].

Following the findings of Danielle and Studies [38], the government might struggle to cope with the developing and increasing demand of the business and economic sphere, if they delay the adoption of new 4IR technologies. Hence, the posturing of the government of any country on policies guiding construction is very important for the acceptance, implementation, and adoption of 4IR technologies. In a case study conducted in South Africa by Danielle and Studies [38], it was revealed that some governments may not be able to sustain current levels of public investment if solid policies guiding the legal and implementation frameworks for 4IR technologies are not swiftly put in place. These findings will surely cause increments in the level of implementation and adoption of 4IR technologies in the long term. Developments in the ICT sector of Africa have been fundamentally determined by increasing mobile digital financial services. The region had virtually half of the world-wide mobile financial records in 2018 and will see the most excellent development in mobile

investments by 2025. Failure to recognize and capitalize on Fourth Industrial Revolution opportunities, equally, will incur substantial hazards for African stakeholders. With no efforts to go further than the obtainable models of entrepreneurship, innovations, and digital development on the continent, African businesses risk falling behind, worsening the universal "digital split" and subordinating their worldwide competitiveness. Even though the current knowledge concerning 4IR is still prehistoric and unfamiliar to many sectors, 4IR is, at present, a buzzword and gaining a toehold across different sectors of the economy [39]. Consequently, 4IR, which is professed as a mixture of many technologies and is supposed to smear the borders between the digital, physical, and biological specialties, is currently attracting growing consideration from academics, business practitioners, and policymakers. Whereas the conception is growing in significance and importance across diverse sectors, there is no agreement on what it involves, and no precise definition has been established to demonstrate its characteristics, despite its continuation since the start of the 21st century [40,41]. Many revolutions and evolutions have been spurred on by Research and Development in a quest to make life easier, and the world friendlier and more habitable. Scientists have been extremely creative and innovative over the past few years in developing new technologies and improving existing ones. From governments of developed and developing countries allocating huge funds to research and education to big multi-national companies spending millions and billions of dollars on product and machinery research, a great impact has been made on the emergence of 4IR. The existing literature has shown that, apart from the industrialized sector, Industry 4.0 has acted as a catalyst for the transformation in other industries [42,43]. For instance, Lasi et al. [44] argued that the introduction of Industry 4.0 law has transformed how the financial sector operates. Through the use of blockchain, a centralized ledger driven by the Fourth Industrial Revolution, Industry 4.0 transformed the financial sector [45]. Blockchain technology eradicates mediators by generating a cryptographically protected trusted ledger linked through an electronic system that authenticates a transaction before its recording. In the same vein, Thuemmler and Bai [46] argued that the healthcare system has likewise benefited from Industry 4.0, because it has allowed for the enhancement of modern diagnostic techniques and the sequencing of genes. Regrettably, there are few texts concerning the profit of technologies powered by Industry 4.0 in the building industry. Consequently, Ngowi et al. [47] and Alwan et al. [48] posited that the absence of previous works concerning the advantage of implementing technologies powered by Industry 4.0 could be connected to poor implementation of new technologies by the building industry.

Hence, the construction industry can follow the manufacturing industry in learning and using 4IR technologies. The amount of information readily available to professionals and individuals alike about the new technologies will go a long way to determining their adoption in the building industry. People will likely not use or adopt something they know less about. The exponential change speed of the latest Fourth IR technology conveyed challenges commonly experienced as multifaceted and even menacing by bona fide estate and building industry practitioners [49]. At present, faced with an Industry 4.0 in which the lives of the population are made more comfortable and sustainable through endless generation of the latest tech-driven services and additional values, seasoned building certifiers might feel plagued as other disciplines impinge on their sphere with tech-savvy "smart" solutions. The expediency, optimism, and self-assurance of millennials may help our industry address the simmering problems we presently have with a sector that is underperforming owing to the overall economic downturn, especially in this region of Africa. Building experts will be better prepared to compete with their counterparts inside and outside of the building sector in this era and age if they have sufficient information about these technologies. The rule of digital twins can be applied to the design and manufacturing of modular apparatus. The law of digital twins is very much useful to scrupulously estimate the design and to assist the designer in avoiding design errors and imperfections resulting from measurement [50].

The idea behind the modular approach is to create reusable, parameterized components that are similar to physical entities beforehand. Apart from the age of the physical object, the components carry out equivalent changes prompted by parameters and combine to form a digital twin model of a real-world industrial unit. The modeling process takes a great deal less time with the modular approach. Owing to its ability to make the same change if the industrial unit design is changed, the digital twin is flexible. This digital twin can quickly certify other potential design solutions through several experiments, in addition to quickly authenticating recent designs.

In geotechnics and metrology, 4IR can be applied for field testing. Field testing goes a long way to ensuring soil stability and finding strength in construction science. This is in agreement with Gogolinskiy et al. [51], who took a broad view of the current conditions and trends in the development of engineering tools and metrology from the specific perspectives of non-destructive testing (NDT) and condition monitoring (CM) at the cusp of the Fourth IR. Thus, global restructuring of the socio-economic and industrial structure of the global economy is required. This has explained the key technological commands of the Fourth Industrial Revolution, analyzed issues in the metrology and equipment-making sectors, and has granted managerial and technological support. One of the reasons for 4IR innovations finally being implemented and seen everywhere is the additional motivation of its relevance to field testing in metrology. The arrival of 4IR has been seen in building professionals' influence on the precision, speed, and uniqueness of 4IR innovation to repeatedly generate or produce work agendas.

The optimal use of building information modeling (3D, 4D, and 5D BIM) and digital twins make it possible to conduct project scheduling automation. Studies have shown that these innovations can go a long way to reducing the time spent on project scheduling. Construction costs and financing can be better managed, planned, and controlled with the application of 4IR innovation. Accordingly, Liu et al. [52] revealed that the workflow design for construction costs can be perfected to effectively manage costs using BIM. Other concepts or innovations in 4IR are being harnessed to create virtual building prototypes (digital twins, data science, virtual reality, augmented reality, and artificial intelligence), which will enhance accuracy during construction. This will, in turn, reduce waste and the cost of maintenance in the long run. It is expected that construction professionals will be encouraged to adopt 4IR innovations for the cost-performance benefits. The budget for overall construction can be accurately estimated using 4IR innovations, leaving zero room for errors. BIM has been implemented at various steps of building and various managerial levels to assist in joint work. Successful teamwork in any framework needs the successful implementation of and devotion to three major components: population, technology, and process. These three components are complementary and synergistic [53]. In the same vein, other innovations in 4IR have been tapped into. Research has also shown the interoperability and compatibility of using these technologies. For instance, when predicting future events based on previous happenings, different concepts are synergized to achieve this. In this case, artificial intelligence and data science are applied to data collected through IoT sensors. These technologies will enhance construction professionals' ability to visualize and conceptualize every stage and aspect of construction projects before the construction phase even begins. This is even more germane in this era of the COVID-19 pandemic, where movements and physical contact are largely restricted. Professionals can lean on this quality of 4IR innovations to effectively carry out their professional duties without interruptions. Another germane driver of the adoption of 4IR innovation that should be given adequate attention is training and education on how to use these technologies. If these technologies are to be adopted and used by construction professionals, the construction industry in the diaspora needs to have a carefully drafted plan to teach and instruct stakeholders in the application and usage of these technologies. Table 1 illustrates the drivers of 4IR in the construction industry.

**Table 1.** 4IR drivers in the construction industry.

| Code | Drivers | References |
| --- | --- | --- |
| D1 | Level of awareness of new technology | [41,54,55] |
| D2 | Training and guidance about the usage of the technology in that area. | [30,31] |
| D3 | High cost of using technology | [52] |
| D4 | The sustainable use of the technology | [34,40,54–56] |
| D5 | Government policy on new technology | [55] |
| D6 | Willingness of professionals to adopt new technology | [42,47,53,55] |
| D7 | Global market or economy | [34,40,55] |
| D8 | Funding Research and Development | [52] |
| D9 | Easy learning ability and usage of new technology | [41,43,53,56] |
| D10 | Mass production using pre-fabrication | [34,40,54,56] |
| D11 | Sufficient information about new technology | [41,54,55] |
| D12 | Designing and manufacturing of modular equipment | [47,55] |
| D13 | Financial feasibility study | [30,31] |
| D14 | Field testing | [52] |
| D15 | Willingness to take risks | [55] |
| D16 | Automation of schedule/register generation | [54,55] |
| D17 | Adoption of prior IT and telecommunication technologies | [47,55] |
| D18 | Incorporate health and safety into the construction process | [42,47] |
| D19 | Cost planning | [31] |
| D20 | Documentation/specification management | [34,40,55] |
| D21 | Collaborative working | [52] |
| D22 | Integration and coordination | [47,55] |
| D23 | Standardization and streamlining | [53,55] |
| D24 | Information availability and sharing | [34,40,55] |
| D25 | Visualization | [30,31] |
| D26 | Selection of important activity to be automated | [55] |
| D27 | Conceptual design of new technology | [52] |
| D28 | Adequate training on usage | [41,54,55] |

## 3. Development of Model and Method of Research

There are twenty-eight (28) drivers produced and deemed suitable to put into practice in 4IR based on the prior work on drivers of 4IR, as shown in Figure 1. Therefore, after delivering a file of 4IR drivers to housing building professionals with the necessary industry know-how, feedback (questionnaire) for the research was obtained. After analyzing these factors and their groups using Exploratory Factor Analysis (EFA), the completeness and accuracy of the 4IR drivers in the grouping were explored.

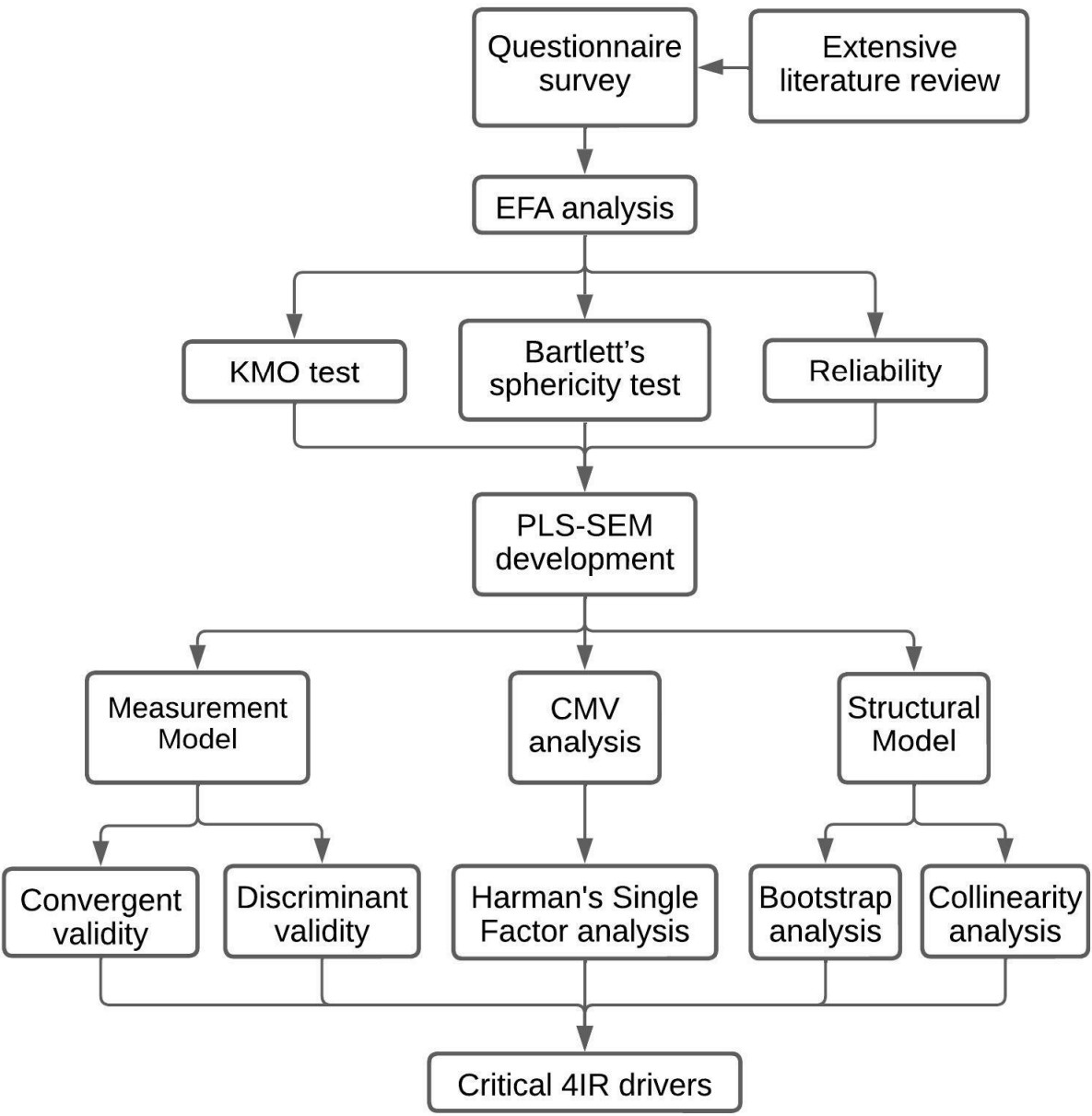

**Figure 1.** Research design.

Partial Least Square Structural Equation Modeling (PLS-SEM) has attracted considerable attention across many areas, predominantly business studies and social sciences [57]. The PLS-SEM procedure has recently been published in well-known SSCI journals and has been a powerful analytical tool used by different researchers [58–60]. The most recent software version, which was used to estimate the data obtained for modeling the priority of the drivers of 4IR using structural equation modeling, is PLS-SMART 3.27. PLS-SEM. It was first recognized for its superiority and predicting capabilities when compared to covariance-based structural equation modeling (CB-SEM) [61]. Even though the contrast between the two procedures is relatively minor [62], this study's statistical analysis included the modeling and evaluation of structural measurement methodologies.

### 3.1. Common Method Variance (CMV)

The common method variance technique was used to calculate the common method bias (CMB). The CMB helps to highlight the inconsistencies (or defects) in an analysis' results that can be attributed to the measurement technique in place of the structures represented by the measurements [63]. The CMB can also be interpreted as a discrepancy

overlap that could be linked to measurement procedures [63]. When data are obtained from a clear basis, such as a self-administered questionnaire, CMV is often tedious [64,65]. In normal conditions, the self-reported data can blow up or avoid the level of examined correlations and, consequently, prompt problems [65,66]. Given that every piece of data for this study is self-reported, subjective, and derived from a single source, it may be essential to find any common method variance, since it is crucial to focus on these issues. Harman's (1976) experiment was used as the basis for a rigorous systematic one-factor test [67]. From the factor analysis, one factor emerged which explained the majority of the variation [65].

### 3.2. Dimension Model and Convergent Validity

The current relationship between the variables and their essential underlying structure is shown by the measurement model [68]. In the subsections that follow, the convergent and discriminant validity of the dimension model are carefully discussed. In addition, convergent validity (CV) is the degree to which two or more measurements (CSFs) of the same construction are consistent (group) [69]. It has been considered to be a subset of the construct's validity. In PLS Cronbach's alpha ($\alpha$) composite reliability scores ($\rho_c$), and the average variance extracted (AVE), are used to determine the convergent validity of the estimated constructs [70]. Nunnally and Bernstein [71] demonstrated a $\rho_c$ value of 0.7 as the threshold of 'modest' composite reliability. For any kind of research, values above 0.70 and 0.60 for exploratory research were deemed acceptable [72]. Suitable convergent validity is shown by values of more than 0.50, which is a common measure used to assess the convergent validity of model constructs [72].

### 3.3. Discriminant Validity and Analysis of Structural Models

Discriminant validity emphasizes the empirically unique nature of the phenomenon being estimated and contends that any measurements that fail to distinguish the phenomenon under SEM are invalid [73]. Campbell and Fiske [74] argued that for discriminant validity to be demonstrated, the similarity across measurements should not be too high.

This research aims to predict the precedence of the 4IR drivers using SEM. Thus, it requires the determination of the path coefficients amongst the measured coefficients. The one-way causal relationship (or path relation) between £ (i.e., drivers of 4IR components) and $ (drivers of 4IR implementation) is shown in Figure 2. The structural connection between the formulas for £, μ, and €1, also known as the inner relation, can be shown as the following linear equation [75]:

$$\mu = \beta\pounds + \text{€}1 \tag{1}$$

where the residual variance at this structural level meant to reside in (€1) and (β) is the route coefficient tying the 4IR components together. The weight (β) of a multiple regression model, in this instance, is the standardized regression weight. Its sign should be statistically significant and concur with the model's predictions.

The major task is determining the importance of the path coefficient. Similar to CFA, the SmartPLS3.2.7 software's bootstrapping technique was used to assess the standard errors of the route coefficients. Based on a recommendation by Henseler et al. [57], 5000 subsamples were required. It then explained how t-statistics are employed in hypothesis testing. A total of four structural equations were developed for the PLS model to capture the intrinsic connections between Equation (1) and the 4IR components.

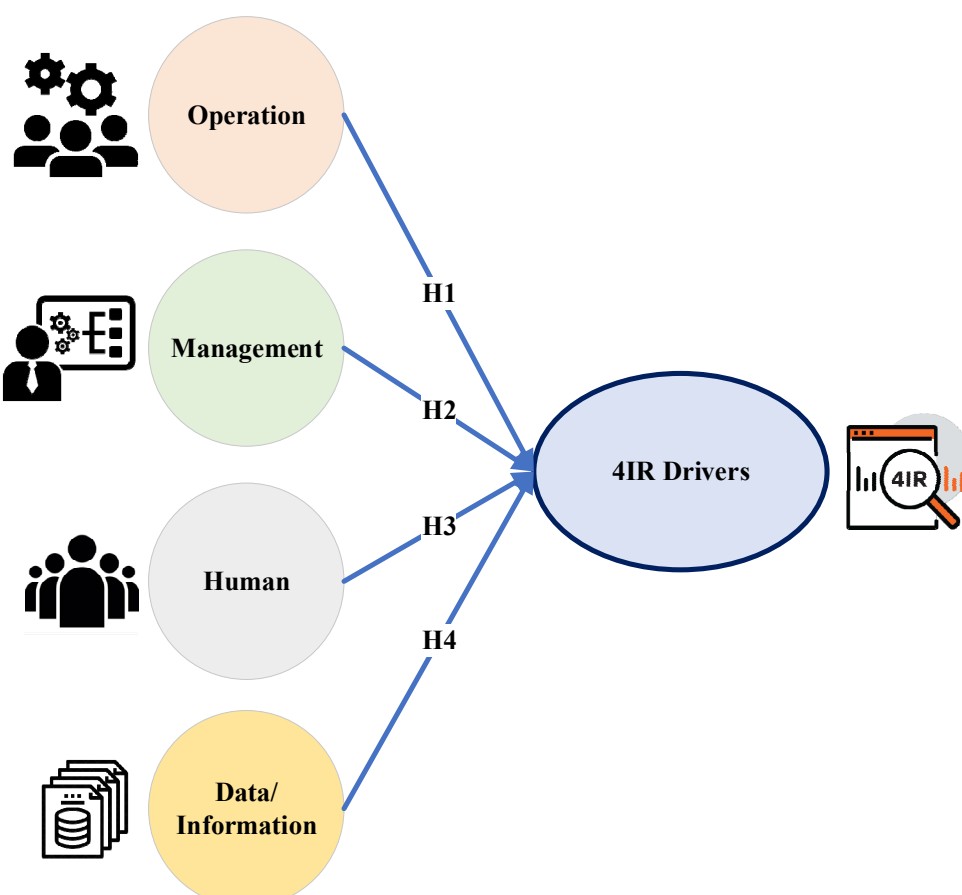

**Figure 2.** Research conceptual framework.

## 4. Data Collection

To understand the driving forces behind 4IR, a larger number of potential residential building sector participants in Nigeria were interviewed using a questionnaire. The three primary elements of this survey were the respondents' demographic profiles, the drivers of 4IR (Table 1), and open-ended questions, which enabled respondents to add any driver(s) that they considered to be essential inclusions.

Three fundamental groups were contacted: clients, consultants, and contractors. Architects, electrical engineers, quantity surveyors, and structural and mechanical engineers might all be added to the list of professions and occupations above. Respondents evaluated 4IR drivers based on knowledge and experience using a Likert 5-point scale, where 5—very high, 4—high, 3—average, 2—low, and 1—no or very low. The Likert scale has been extensively applied in the literature [76–83]. However, in Nigeria, 4IR is incredibly new; therefore, stratified sampling of the particular subpopulation employed in this study was taken into consideration [29].

The sample size should be based on objective methodological and statistical analysis [84], and Chandio [85] agreed that the more complex the statistical analysis seems, the greater the sample size required. This means that for every study, the sample size is related to the chosen statistical approach [84]. Consequently, sample size requirements were derived according to the chosen statistical analysis technique to generate the 4IR implementation model. SEM was selected for this study; accordingly, the sample size, this should be sufficient to accomplish high statistical power to discard alternative models [86]. In this situation, Krejcie and Morgan [87] confirmed that the maximum sample size could be suitable for developing SEM.

On the other hand, many other researchers oppose maximization and recommend optimizing the sample size [88]. They argue that it is not cost-effective and time-efficient at

a certain level; however, another additional advantage of extra numbers is that they are minimal. Therefore, the minimum measured sample size to achieve the desired statistical power level was given in SEM [89,90]. SEM, like other statistics, requires a suitable sample size in order to acquire consistent estimations [91]. Gorsuch [92] suggested a minimum of five participants for every construct and 100 individuals for every data analysis. To provide a robust SEM, Harris and Schaubroeck [93] recommend a sample size of 200 to generate SEM. Kline [88] also recommended a sampled size of 200 for complex path models, while Yin [94] also recommended a sampled size somewhere above 100, but more preferably above 200. This is also agreed by Hair Jr et al. [91], who advocate for at least 200 sample sizes. Furthermore, the analysis of SEMs is also too sensitive to the extent that any difference can easily be identified when the sample size surpasses 400 to 500 participants, which shows poor fit [90]. Due to the SEM approach used in this study, a total of 257 of the 348 participants were personally contacted (through self-administration of the questionnaire), translating to a response rate of roughly 73.85%. This rate of return was deemed suitable for this type of research [95,96].

## 5. Data Analysis and Results

### 5.1. Exploratory Factor Analysis

Twenty-eight (28) elements associated with 4IR drivers underwent an exploratory factor analysis (EFA) to determine their factorability. Several well-known factorability parameters were used for the connection. Kaiser–Meyer–Olkin (KMO) factor homogeneity assessment has been widely used to confirm that the preferential correlations between variables are kept to a minimum [97]. Table 2 shows the sampling strategy's suitability in terms of the quantity of the data set and the size of the population. According to Tabachnick et al. [98], Bartlett's Test of Sphericity must be considered to be appropriate ($p = 0.05$). It has been also suggested that factor analysis is suitable for KMO index values greater than 0.60. The KMO index ranges from 0 to 1. Therefore, the closer it is to 1, the better. Factor analysis is applicable in both of these cases because the KMO index, in this case, is 0.813 and the Test of Sphericity has a significance value of 0.000. Table 2 displays the results of the KMO and Bartlett's Tests.

**Table 2.** KMO and Bartlett's Tests of the drivers for the adoption of 4IR.

| Sampling Adequacy measure according to Kaiser–Meyer–Olkin | | 0.813 |
|---|---|---|
| Bartlett's Test of Sphericity | Approximate Chi-Square | 4381.906 |
| | df | 378 |
| | Sig. | 0.000 |

It is appropriate to include each variable in the component analysis since the total diagonals of the anti-image correlation matrix are bigger than 0.5. The initial communities are forecasts of variance for every variable that all the components have considered. Small values (below 0.3) indicate variables with a poor factor solution fit. The criterion for this inquiry is exceeded in each of the initial communities. The significance of each loading factor is above 0.5. The EFA analysis was conducted comprising all 28 questions of the survey tool. The findings reveal four factors with eigenvalues greater than 1. As indicated in Table 3, the six components' eigenvalues and total variance constitute 58.646%. However, identification of the four (4) separated components becomes crucial to allow for a clear and succinct interpretation. There is little to no established designation convention for factor analyses' retrieved components in the literature. Consequently, the implication is that labeling these elements may be subjective and dependent on the viewer's or researcher's perception, the analyst's background, and training/education. Following thorough deliberation over the proper classification procedure, the following names were chosen; (i) operation drivers, (ii) management drivers, (iii) human drivers, and (iv) data/information drivers.

**Table 3.** Rotated component matrix of the drivers for adoption of 4IR.

| Drivers | Components | | | |
|---|---|---|---|---|
| | **F1** | **F2** | **F3** | **F4** |
| **Operation Drivers** | | | | |
| D12 | 0.894 | | | |
| D13 | 0.852 | | | |
| D9 | 0.833 | | | |
| D11 | 0.809 | | | |
| D10 | 0.766 | | | |
| D1 | 0.752 | | | |
| D19 | 0.676 | | | |
| D14 | 0.649 | | | |
| D8 | 0.611 | | | |
| D6 | 0.553 | | | |
| D16 | 0.531 | | | |
| D28 | 0.442 | | | |
| **Management Drivers** | | | | |
| D17 | | 0.786 | | |
| D4 | | 0.670 | | |
| D5 | | 0.656 | | |
| D22 | | 0.639 | | |
| D18 | | 0.610 | | |
| D7 | | 0.542 | | |
| D21 | | 0.524 | | |
| D23 | | 0.455 | | |
| D20 | | 0.448 | | |
| **Human Drivers** | | | | |
| D3 | | | 0.725 | |
| D2 | | | 0.651 | |
| D27 | | | 0.582 | |
| D15 | | | 0.466 | |
| **Data/Information Drivers** | | | | |
| D26 | | | | 0.716 |
| D25 | | | | 0.471 |
| D24 | | | | 0.406 |
| **Eigenvalues** | **6.607** | **3.953** | **3.840** | **2.605** |
| **Total Variance Explained (%)** | **24.158** | **16.453** | **10.866** | **7.169** |
| **Method of Extraction:** Principal Component Analysis. | | | | |

"Operation drivers", which were the first component extracted and contain twelve (12) items, depict drivers that are correlated with processes involved in the operations and procedures of the new technologies. The twelve items are as follows: the designing and manufacturing of modular equipment, with a significance of 0.894; a financial feasibility study, with a significance of 0.852; easy learnability and usage of new technologies, with a significance of 0.833; sufficient information about new technology, with a significance of 0.809; mass production using pre-fabrication, with a significance of 0.766; a level of awareness of new technology, with a significance of 0.752; cost planning, with a significance of 0.676; field testing, with a significance of 0.649; funding Research and Development, with a significance of 0.611; willingness to take risks, with a significance of 0,553; the automation of schedule/register generation, with a significance of 0.531; and adequate training in usage, with a significance of 0.442.

The name propounded for the second extracted component is "management drivers". Management drivers refer to activities such as policies, organizations, and individuals that could potentially govern the smooth sailing of these technologies. The items loaded under this component total nine (9) and they include: the adoption of IT and telecommunications technologies, with a significance of 0.786; the sustainable use of the technology, with a

significance of 0.670; government policy on the new technology, with a significance of 0.656; integration and coordination, with a significance of 0.639; incorporating health and safety into the construction process, with a significance of 0.610; the global market or economy, with a significance of 0.542; collaborative working, with a significance of 0.523; standardization and streamlining, with a significance of 0.455; and documentation/specification management, with a significance of 0.448.

"Human drivers" is the name allotted to the third extracted component. As the name implies, the factors represent variables that seemingly depict human inputs in the adoption of these new technologies. It contains four (4) items, and these include the high cost of using the technology, with a significance of 0.725; training and guidance in the usage of new technology, with a significance of 0.651; the conceptual design of new technology, with a significance of 0.582; and the willingness of professionals to embrace new technology, with a significance of 0.466.

Furthermore, statistics on reliability were created for the factors obtained through the EFA. To determine the variables for each phase of the factor, the highest loading of each variable in the structural matrix was employed. According to Nunnally [99], a Cronbach's alpha value greater than 0.6 is deemed suitable for newly produced measurements. Those over 0.75, on the other hand, are seen to be tremendously accurate when the usual value is 0.7. As a result, since the Cronbach's alpha values were higher than 0.6, the results were deemed appropriate. All of the objects' set average correlations are greater than 0.3, indicating stable internal variables [100].

Following the factor analysis, as shown in Figure 2 the conceptual model for the study was created, and it contains the four hypotheses listed below:

- $H_1$: Operation positively influences 4IR drivers.
- $H_2$: Management positively influences 4IR drivers.
- $H_3$: Humans positively influence 4IR drivers.
- $H_4$: Data/information positively influence 4IR drivers.

### 5.2. PLS-SEM Model Analysis

#### 5.2.1. Common Method Bias

Common method bias (CMB), which is a measurement of errors, affects a study's validity (variance). It represents systematic error variance concerning the measured and estimated variables [101]. It can be evaluated using Harman's single-component model analysis, which may be an indication of various structural metrics [67]. The variance of the conventional approach was assessed in this study using the single-factor test [102]. The CMB has no impact on the data if the components' combined variance is less than 50% [67]. The results show that the first set of components accounts for 22.75% of the total variance, showing that the common method variance cannot affect the results because it is less than 50% [67].

#### 5.2.2. Model Measurement

For the reflective measurement models (drivers) in PLS-SEM, internal reliability, convergent validity, and discriminant validity evaluations are necessary. Once the measurement model has been proven to be valid and reliable, the structural model will be evaluated [103]. The model's constructs all satisfy the criterion of $\rho_c > 0.70$, as shown in Table 4, and are therefore acceptable [104].

**Table 4.** Convergent validity's output.

| Constructs | Cronbach's Alpha | Composite Reliability | AVE |
|:---:|:---:|:---:|:---:|
| Operation | 0.932 | 0.944 | 0.678 |
| Management | 0.934 | 0.945 | 0.655 |
| Human | 0.905 | 0.934 | 0.779 |
| Data | 0.842 | 0.904 | 0.758 |

The results in Table 4 also demonstrate that all the constructions passed the AVE test. For AVE values to be acceptable, they are required to be above 0.5 [70]. All of the components in this study have estimated AVE values >0.5 (Table 5). These are greater than 50% according to the PLS algorithm 3.0. These results further demonstrate the measurement model's internal convergence and consistency. They likewise show that the measurement components do not assess any other constructs within the research model and are effectively measured for each construct (group). High outer loadings on a construct show that the constituent parts of each construct are closely related. The basic guideline is that items with extremely low external loadings (below 0.4) must be eliminated from the scale [62]. The external loadings of the initial and modified measurement models are shown in Figure 3 for each item. As a result, all exterior loads from the initial measurement model have been removed, except for three variables, i.e., the "D12", "D14", and "D6" items. Their minimal contribution to the pertinent constructs is shown by the omission, which is the result of a low loading factor of less than 0.5.

As soon as a construct differs significantly from other constructs based on observable standards, discriminant validity tends to be well-defined. Consequently, the construct's founding discriminatory validity contends that it is distinct and may account for occurrences that other constructs in the model are unable to adequately characterize [105]. The Fornell–Larcker (1981) criterion and the HTMT (Heterotrait–Monotrait ratio of correlations) criterion are two discrete techniques for assessing discriminant validity. It is possible to assess the discriminant validity by comparing the AVE's square root of each construct to the correlations of a given construct with any other constructs. Given that the correlation between the latent variables is calculated using the square root of the AVE, it needs to abide by Fornell and Larcker [70] principles. Table 5 shows that the result has confirmed the discriminant validity of the measurement model [106].

However, several scholars disagreed with Fornell and Larcker [70]'s criterion for distinguishing discriminant validity. Consequently, Henseler et al. [107] suggested the use of the Hetrotrait–Monotrait (HTMT) criterion ratio, as an additional tool for evaluating discriminant validity. HTML is a different method for evaluating the discriminant validity of variance based on SEMs and calculates the exact correlation between the two constructs under ideal measurement conditions, i.e., if they are unfailingly reliable and error-free. The discriminant validity in this study was likewise evaluated using the HTMT model. Hair et al. [73] suggested that for two constructs to be considered independent, the HTMT values have to be between 0.85 and 0.90 or lower. If the model's constructs are conceptually different from one another, the HTMT values have to be lower than 0.85 and lower than 0.90, respectively. The HTMT values for each construct in this study are displayed in Table 6. The constructs demonstrated sufficient discriminant validity.

**Table 5.** The result of the Fornell–Larker criterion.

| Construct | Data | Human | Management | Operation |
|---|---|---|---|---|
| Data | **0.871** | | | |
| Human | 0.349 | **0.883** | | |
| Management | 0.27 | 0.308 | **0.809** | |
| Operation | 0.384 | 0.326 | 0.194 | **0.823** |

**Table 6.** The result of the Heterotrait–Monotrait ratio (HTMT).

| Constructs | Data | Human | Management | Operation |
|---|---|---|---|---|
| Data | | | | |
| Human | 0.383 | | | |
| Management | 0.288 | 0.333 | | |
| Operation | 0.427 | 0.351 | 0.203 | |

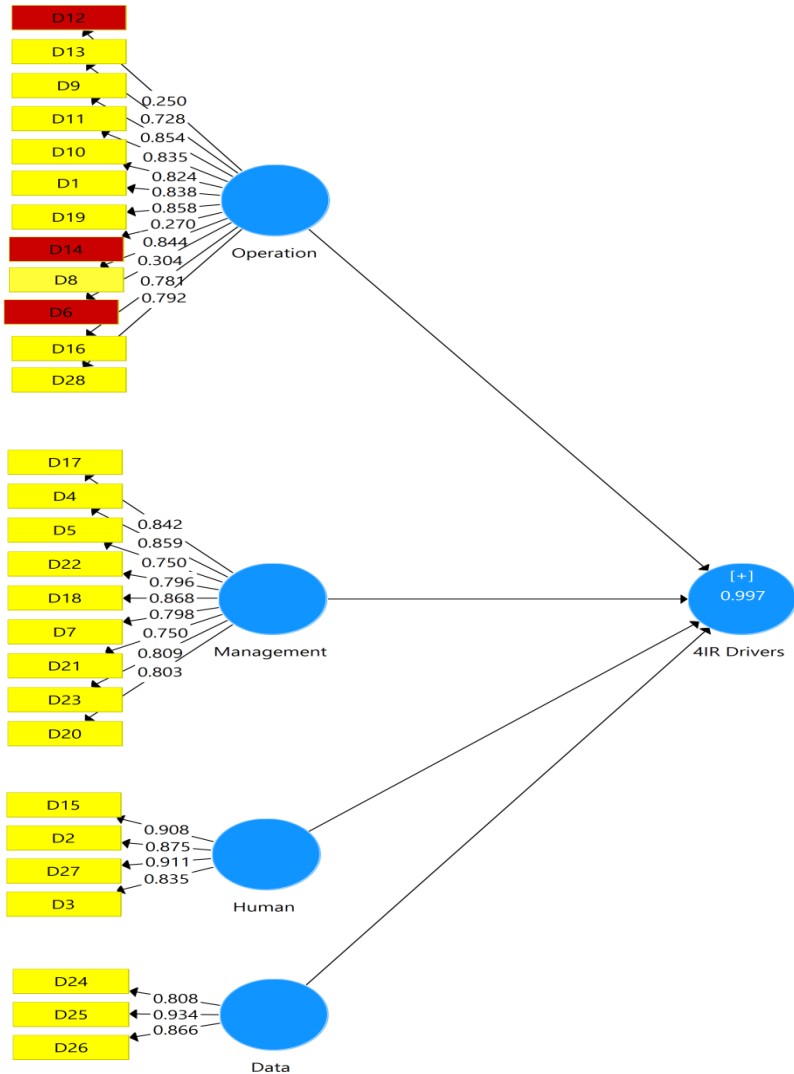

**Figure 3.** PLS-SEM model.

5.2.3. Path Model Validation

Once it is established that the drivers of 4IR are formative constructs, it is possible to study the collinearity between the constructs' formative objects by determining the value of the variable inflation factor (VIF). All VIF values are markedly below 3.5, which indicates that these subdomains of each variable contributed to the higher-order structures in a specific manner. Additionally, a bootstrapping tool was used to forecast the importance of the path coefficients. According to Table 7, all paths are statistically significant at the ≤0.01 level [69].

**Table 7.** Hypotheses, SD, and *p* values for the model.

| Paths | β | SD | *p* Values |
|---|---|---|---|
| Operation → 4IR Drivers | 0.508 | 0.055 | 0.000 |
| Management → 4IR Drivers | 0.620 | 0.056 | 0.000 |
| Human → 4IR Drivers | 0.247 | 0.027 | 0.000 |
| Data → 4IR Drivers | 0.210 | 0.004 | 0.000 |

## 6. Discussion

Following the devastating impacts of the coronavirus pandemic (COVID-19) (Olanrewaju et al., 2022), the application of 4IR needs to be embraced by the construction industry.

This study revealed that the 4IR drivers for sustainable residential building include four categories (operation drivers, management drivers, human drivers, and data drivers). 4IR is a new dimension, and many industries are converting to its application since it aids in responsive decision-making and quality strategy implementation [108].

Operation is critical, and it positively influences 4IR drivers ($\beta = 0.508$, SD = 0.055, $p = 0.000$), as revealed by the PLS-SEM result. This category of drivers contains twelve drivers (the designing and manufacturing of modular equipment; a financial feasibility study; easy learnability and usage of new technologies; sufficient information about new technology; mass production using pre-fabrication; a level of awareness of new technology; cost planning; field testing; funding Research and Development; a willingness to take risks; the automation of schedule/register generation; and adequate training in usage) and accounted for 24.158% of the total variance within 4IR drivers for sustainable residential building. Hence, $H_1$ was accepted because the impact is significant. It is the impact category with the second largest impact on 4IR drivers. It has emphasized the need to pay attention to the drivers in this category. Oke and Fernandes [39] showed the importance of 4IR technologies such as data analytics and artificial intelligence in education. However, it was observed that the sector is not fully ready for 4IR. Consequently, Kafile [109] revealed a strong demand for the use of emerging technologies for project management to maintain a competitive advantage in the digital business world. The findings from previous studies revealed that there is a need to increase the uptake of emerging technologies in the construction industry with a view to enhancing project delivery to meet the high demand of the increasing population. Innovative 4IR technologies are capable of enhancing project delivery operations and preventing project failure.

Management is another vital component of 4IR, and it positively influenced 4IR drivers ($\beta = 0.620$, SD = 0.056, $p = 0.000$), as revealed by the PLS-SEM result. This category of drivers comprised nine drivers (the adoption of prior IT and telecommunications technology; the sustainable use of the technology; government policy on the new technology; integration and co-ordination; incorporating health and safety into the construction process; global market or the economy; collaborative working; standardization and streamlining; and documentation/specification management) and accounted for 16.453% of the total variance within 4IR drivers for sustainable residential building. Hence, $H_2$ was accepted because the impact is significant. The management category has the largest impact on 4IR drivers. This implies that the management component of 4IR is critical to its adoption. The management of companies must evaluate how 4IR can influence their organization to ensure it poses no threats to fundamental human rights [110]. Thus, Lanteri [108] posited that 4IR could improve collaborative intelligence, which could enhance sustainable decision-making with fewer human efforts. Management is critical to the success of the delivery of residential building projects, and the use of 4IR technologies can optimize the management of construction projects.

Human-related drivers are key determinants of 4IR, and they positively influence 4IR drivers ($\beta = 0.247$, SD = 0.027, $p = 0.000$), as revealed by the PLS-SEM result. This category comprised four drivers (training and guidance about the usage of new technology; the conceptual design of new technology; and the willingness of professionals to adopt new technology) and accounted for 10.866% of the total variance of 4IR drivers for sustainable residential building. Hence, $H_3$ was accepted. This category of drivers has a low impact compared to the management and operation categories. Conversely, it is still a crucial component of 4IR drivers because human factors play a vital role in adopting emerging technologies. Nath [111] emphasized that 4IR technologies must be human-centric to transform 4IR integrated sustainable development. Yang et al. [112] posited that the relationship between humans and smart devices is one of the fundamental research hotspots in 4IR. However, Malik et al., (2020) argued that the ethical, moral, and legal implications of 4IR technologies, such as the implementation of artificial intelligence by organizations, have not been largely considered in the existing literature despite their huge impact after adoption. Taiwo and Vezi-Magigaba [113] stress the need for employers to invest heavily

in human capital development with a view to enhancing 4IR adoption. They implied that humans are a critical part of 4IR implementation for the construction industry and cannot be neglected.

In addition, data are crucial to 4IR, and they positively influence 4IR drivers ($\beta$ = 0.210, SD = 0.004, $p$ = 0.000), as revealed by the PLS-SEM result. This category of drivers comprised three drivers (the selection of important activity to be automated; visualisation; and information availability and sharing) and accounted for 7.169% of the total variance within 4IR drivers for sustainable residential building. Hence, $H_4$ was accepted. It had the lowest impact on 4IR drivers; however, it is still an essential part of the 4IR network as it powers digital business models [108]. Data constituted the heart of 4IR, combining advances in robotics, the Internet of Things (IoT), artificial intelligence (AI), 3D printing, quantum computing, genetic engineering, and other technologies [114]. Data are an essential part of human life, and the construction industry has generated a wide range of data at every phase of the building life cycle, which has made them an integral part of 4IR. Studies have proposed solutions to improving data security and sharing [115,116]. For instance, Ray and Bagwari [115] proposed a solution that protects consumers' data, which boosts their confidence in smart home technologies. Xu [116] also proposed a blockchain-based solution for a smart public safety system that enhances traceability, secure data sharing, and operations among participants. Previous studies have also highlighted the benefits of visualization through emerging technologies such as building information modeling (BIM) and virtual reality in the construction process, as it enhances sustainable decision-making [117]. This shows that visualization plays a vital role in the building development process and enhances the quality of decisions made. Given the findings from existing studies, it can be deduced that 4IR technologies can help enhance the data systems of sustainable residential buildings. A huge volume of data are generated during the construction process, and 4IR technologies such as blockchain, IoT, and BIM can be used to collect, store and present data. These data can be used to improve the performance of residential buildings. For instance, IoT can be used to obtain data regarding the environmental performance of the building, while blockchain can be used to store data generated from the building.

## 7. Conclusions

The literature is concordant about the significance of 4IR adoption in the building industry. The growth of the construction industry depends on 4IR, which is a critical tool in hastening the recovery of the industry from the adverse impacts of the coronavirus pandemic. This study revealed the critical drivers of 4IR for sustainable residential building development. These drivers were categorized into four typical categories (operation, management, human, and data/information) through factor analysis. The study further revealed that management and operation are the two driver categories which strongly influence 4IR adoption in the construction industry. As a result, the adoption of 4IR technologies for sustainable residential development should concentrate on the management and operation dimension of 4IR drivers in order to enhance building performance. To demonstrate the effects of the driver categories on 4IR, this study suggested a special empirical model utilizing PLS-SEM. This model aims to help stakeholders in the construction industry improve 4IR adoption.

### 7.1. Implications and Contributions

The PLS-SEM model highlights the significance of the four 4IR driver categories for sustainable residential building development. The significant drivers of 4IR are presented in the model. It is anticipated that 4IR technologies will improve the performance of residential building projects, which is required as the world population continues to increase sporadically. For instance, 3D and 4D printing technologies can be used to enhance the delivery of sustainable residential building projects with an effective balance between cost, time, quality, and sustainability. This paper revealed that operation and management drivers are critical to taking the delivery of residential buildings to the next level. These

drivers are of great importance to construction industry stakeholders as they can propel the industry further by facilitating 4IR adoption. They are also important to the government, including policymakers, since they help them make sustainable policies that drive 4IR adoption. In addition, no research on 4IR has categorized and used PLS-SEM to explore the relationships between the 4IR driver categories. Therefore, the mathematical model proposed in this study highlights the critical drivers for 4IR adoption and their significance. Moreover, this research made the following hypothetical contributions to the body of knowledge on 4IR:

- Although several studies have explored the drivers of 4IR in different economic sectors of countries worldwide. This study focused on 4IR for the construction industry, particularly sustainable residential building development. The study presented a well-categorized list of 4IR drivers based on their latent structure significance.
- The PLS-SEM model is the first to predict 4IR drivers for sustainable residential buildings. It is expected that 4IR will aid in the development of sustainable residential buildings and will improve the quality of people's lives. Correspondingly, construction industry stakeholders would likely exploit this model to foster 4IR adoption in the construction industry.

*7.2. Limitations and Potential Research Areas for the Future*

This study has made a substantial and remarkable contribution to understanding the benefits of 4IR adoption in the construction sector. However, there are some gaps in the study's generalization and breadth:

i. The study solely took into account Nigerian construction industry specialists. To better generalize the study results, additional nations must be explored further. Future studies might look into the factors influencing 4IR in other nations.

ii. Secondly, the study did not address how human behavior may affect the uptake of 4IR technology. Using technology adoption models such as the Technology Acceptance Model (TAM), the Unified Theory of Acceptance and Use of Technology (UTAUT) model, the Motivational Model (MM), the Uses and Gratification Theory (U&G), and Diffusion, future studies should examine how construction professionals behave toward emerging 4IR technologies.

**Author Contributions:** Research Idea: A.F.K. Conceptualization, A.F.K. and A.E.O.; methodology, A.F.K. and A.E.O.; software, A.F.K.; validation, All Authors, formal analysis, A.F.K.; investigation, A.F.K. and A.E.O.; resources, All authors; data curation, All authors.; writing—original draft preparation, A.F.K., O.I.O. and A.E.O.; writing—review and editing, All authors.; visualization, All authors.; supervision, A.F.K., O.I.O. and A.E.O.; project administration, All authors. All authors have read and agreed to the published version of the manuscript.

**Funding:** This research received no external funding.

**Institutional Review Board Statement:** Not applicable.

**Informed Consent Statement:** Not applicable.

**Data Availability Statement:** All the data from this study have been presented in the paper. However, further inquiries may be directed to either the first or corresponding authors.

**Conflicts of Interest:** The authors declare no conflict of interest.

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
