# Peer review of "Exploring the 4IR Drivers for Sustainable Residential Building Delivery from Social Work Residential Perspective—A Structural Equation Modelling Approach"

_sustainability, doi:10.3390/su15010468_

Round 1

Reviewer 1 Report

Dear authors, your work seems to me to be methodologically sound, but some improvements and clarifications are required.

1) Give an immersion context to the Nigerian situation (I found it difficult to understand the double author affiliation 1, "&"). I suggest you use the FSI data breadth (https://fragilestatesindex.org/country-data/), it is a comprehensive indicator, well cited and with which the situation in Nigeria has been previously studied.

2) Use more recent literature in the introduction and literature review (Drivers for 4IR Implementation), the world reality of 2021 and 2022 is very different and has strongly impacted the countries' reality and technology implementation.

3) Table 1 requires to be specific to which reference or references each driver corresponds.

4) Provide a clearer wording between lines 312 to 319, what is the sample, referring to Yin is not enough to meet SEM criteria, Yin criteria are general and allow ad-hoc samples, but it does not pass the statistical tests.

5) The results reading is made simpler for the reader if they finish section 3, with a summary table of all the cut-off parameters to be evaluated.

6) The discussion requires a more frontal debate with other authors (articles), in some lines it seems to be reading a simple conclusion. They should strongly emphasize the scientific value of their findings.

7) In 7.2.ii, it is recommended that you incorporate some guiding quotes to your readers. Your article can be the starting point for other researchers.

8) Take care of the journal format, especially citations and references (e.g., line 515 and authors' contributions).

Reviewer 2 Report

The proposed paper is very interesting. 

The authors need to put more effort into writing the introduction and introducing the problem that is being dealt with in the paper. The authors also need to write more about the research gaps and what way the current paper is attempting to fill in those research gaps.

Please complete a few issues:

* The aim of the paper should be more specific.

*Please explain in the paper the extent to which the developed paper can be relevant to the international scientific field.

*In conclusion, please indicate to what extent the proposed paper is innovative and what is its scientific contribution in the realization of Sustainable Residential Building? 

* The study should indicate recommendations in a time horizon. What do the authors think should be the priority in the realization of the  Fourth Industrial Revolution (4IR)?

* The discussion and conclusions section needs to be fleshed out, primarily indicate to more details to 4IR adoption in the construction sector. What activities should be implemented and in what area undertaken.

Round 2

Reviewer 1 Report

The authors have largely made the required improvements.

Although they did not incorporate cut-off parameters to the tables or at least (*) to the highlighted values, nor did they add references to section 7.2.ii. These details differentiate a published article from an article that will be cited in the future.

There are still some form changes, such as references and authors' contribution lists, but these will be required in the final edition, if full endorsement is achieved.

Good luck with the other reviewers.

Reviewer 2 Report

Thank you for the consideration of the comments.  I support the publication of the article.